# The Coexistence of *Blastocystis* spp. in Humans, Animals and Environmental Sources from 2010–2021 in Asia

**DOI:** 10.3390/biology10100990

**Published:** 2021-09-30

**Authors:** Adedolapo Aminat Rauff-Adedotun, Farah Haziqah Meor Termizi, Nurshafarina Shaari, Ii Li Lee

**Affiliations:** 1School of Biological Sciences, Universiti Sains Malaysia, Kuala Lumpur 11800, Malaysia; adedotunameenah@gmail.com (A.A.R.-A.); farahhaziqah@usm.my (F.H.M.T.); 2Kulliyyah of Medicine & Health Sciences, Universiti Islam Antarabangsa Sultan Abdul Halim Mu’adzam Shah, Kuala Ketil 09300, Malaysia; shafarinashaari@unishams.edu.my

**Keywords:** *Blastocystis*, subtypes, epidemiology, one health, Asia

## Abstract

**Simple Summary:**

*Blastocystis* spp. are unicellular parasites that infect the gastrointestinal tract of humans and animals. Their occurrence in the environment had been detected in water sources, thus causing contamination. The presence of the parasites in humans, animals and environmental sources in Asia were reviewed according to countries in Asia, different categories of human and animal populations, and environmental sources including water samples, food and ambient air. The coexistence of the parasites poses a public health concern as the parasites are commonly found in most studies. Hence, there is a growing interest in the study of *Blastocystis* spp. Due to the isolation of *Blastocystis* spp. from living and non-living sources, a collaborative, multisectoral and transdisciplinary approach known as One Health is proposed for future study of *Blastocystis* spp. in order to achieve optimal health outcomes through the recognition of interconnection between people, animals and their shared environment.

**Abstract:**

*Blastocystis* spp. are controversial unicellular protists that inhabit the gastrointestinal tract of humans and a wide range of animals worldwide. This review provides an overview of the prevalence and distribution of *Blastocystis* spp. and their subtypes throughout Asia. Research articles reporting on the presence of *Blastocystis* spp. in locations within Asia, between 1 January 2010, and 10 May 2021, were obtained from Scopus, PubMed, and Google Scholar. In 427 articles, the prevalence of *Blastocystis* spp. in 31 countries within the last decade was revealed. Isolates were found in humans, various mammals, birds, reptiles, insects, water sources, vegetables, and ambient air. Prevalence of *Blastocystis* spp. varied widely across host categories. Subtypes identified throughout Asia were STs 1–14, and ST18–22 (novel subtypes). ST1, ST2, ST3, ST4 were the most frequently isolated in humans; ST5 in pigs; ST10 and ST14 in goats, sheep, and cattle; and ST6 and ST7 in chickens. ST1 and ST3 were most common in water samples. ST1, ST2, ST3, ST4, ST5 and ST6 were shared by humans, animals, and water sources. There is a growing interest in the study of *Blastocystis* spp. and their subtypes in Asia. Due to the isolation of *Blastocystis* spp. from biotic and abiotic sources in Asia, the application of the One Health (OH) approach to the study of *Blastocystis* spp. is proposed for improved perception of this organism.

## 1. Introduction

*Blastocystis* spp. are anaerobic unicellular eukaryotes that are widespread among humans and animals around the world [1,2,3]. They reside in the gastrointestinal tract wherein their role in gut health and disease is unresolved [4]. Several attempts were made at the classification of *Blastocystis* spp. by means of physiological and morphological characteristics [5], however, its place as a member of the phylum stramenopiles was revealed by Silberman et al. [6] based on phylogenetic analysis of the small subunit ribosomal rRNA (SSU rRNA) gene.

*Blastocystis* spp. exhibit morphological and genetic polymorphism [7]. The six morphological forms described in the literature are vacuolar, granular, amoeboid, cyst, multivacuolar and avacuolar [8]; each of these forms show substantial variations in size [9]. Discerning one *Blastocystis* spp. isolate from another by morphological means alone poses a big challenge as isolates from different hosts appear similar [1].

Differences in the nucleotide sequences of the SSU rRNA gene of *Blastocystis* spp. isolates demonstrate the organism’s substantial genetic heterogeneity [10,11]. According to a consensus on the terminology of *Blastocystis* spp. subtypes proposed by Stensvold et al. [10], *Blastocystis* spp. isolates are referred to as *Blastocystis* spp. ‘subtypes’ (STs). These designations are based on the differences among the SSU rRNA gene sequences; and by 2013, 17 different STs (ST1 to ST17) of *Blastocystis* spp. had been acknowledged [11]. Eleven additional STs (ST18 to ST28) have been proposed since then, although the validity of four of these STs (ST18 to ST20, and ST22) are being contested [12]. Infections with *Blastocystis* spp. ST1 to ST9 and ST12 have been reported in humans [13,14]. All *Blastocystis* spp. STs have, however, been widely isolated from non-human hosts, with the exception of ST9, whose first identification in a non-human host was by Noradilah et al. [15] in chickens reared by aborigines of rural Malaysian communities.

*Blastocystis* spp. are transmitted through the fecal-oral route via the ingestion of feces-contaminated food and water, with the cyst form as the only transmissible form [9,14,16]. Molecular epidemiological studies have revealed possible human-to-human, foodborne, waterborne and zoonotic transmission [17,18,19,20,21,22,23,24,25,26,27]. For example, Eroglu and Koltas [19] reported the isolation of *Blastocystis* spp. subtype 1 from *Blastocystis* spp. positive patients, their pets and the tap water they drank from. Likewise, the presence of *Blastocystis* spp. subtype 4 in humans, the animals they reared and the rivers they visited regularly were observed in a rural community in Nepal by Lee et al. [18]. It is also worth mentioning that *Blastocystis* spp. are included as waterborne pathogens in the World Health Organization’s publications on drinking water quality [28], implying possible public health concerns.

Globally, increasing interactions between humans and animals (domestic, livestock, wildlife) at close proximity cannot be overemphasized. The rapid growth perceived in areas of agriculture, urbanization, industrialization, and international travel and trade have all contributed greatly to these interactions [29,30]. A human-animal-environment interface has emerged from the dynamic relationships between humans and animals; a clear understanding of the risks at this interface would allow better public health outcomes [29]. This is the One Health (OH) holistic approach, which considers health in the context of human, animal and environmental relationships [31]. It urges the use of interdisciplinary collaborative effort to attain optimal health for humans, animals, plants, and the environment. Bearing in mind that the role of *Blastocystis* spp. in the host gut, whether as mutualists, commensals, or pathogens, has yet to be ascertained [32]; the study of this organism from an ecological standpoint is required.

Studies abound on the prevalence of *Blastocystis* spp. from around the world revealing the various host groups and geographic distribution of this intestinal protist. The growing use of polymerase chain reaction (PCR)-based approaches has, equally, broadened the understanding of genetic diversity and transmission of *Blastocystis* spp. Over the last decade, *Blastocystis* spp. research in Asia has noticeably intensified. Rauff-Adedotun et al. [33] observed an increase in the studies of *Blastocystis* spp. infection in animals in Southeast Asia over the last decade. This research direction is deemed timely considering the role of agriculture, industrialization and globalization on the rapid economic growth that is taking place in the Asian region; as well as the resulting large and growing human and livestock populations, high levels of interspecies interaction, and large-scale ecological change.

This article serves as a summary of the prevalence of *Blastocystis* spp. and the distribution of its subtypes in humans, animals, environmental, and food sources across Asia in the last decade.

## 2. Materials and Methods

Articles on *Blastocystis* spp. research carried out within the continent of Asia were searched for in three electronic databases: Scopus, PubMed, and Google Scholar. The search covered articles published between 1 January 2010, and 10 May 2021. Duplicate articles from the three databases were removed; experiental studies, case reports, review articles, articles that did not report a positivity percentage and articles with unclear/confusing information were also excluded. Articles on the prevalence/occurrence and/or subtypes of *Blastocystis* spp. in both life and non-life sources undertaken within Asia were selected. The information extracted from each article included country of study, method(s) of detection of *Blastocystis* spp., host(s) of study, number of samples examined, number of samples positive, subtypes identified with corresponding numbers of isolates, author(s) and publication dates. Studies were retrieved on *Blastocystis* spp. in humans, various animal hosts, water sources, vegetables, and ambient air.

A total of 427 manuscripts met inclusion criteria, these studies were for 31 Asian countries/regions (Bangladesh, Cambodia, China, India, Indonesia, Iran, Iraq, Israel, Japan, Jordan, Korea, Laos, Lebanon, Malaysia, Myanmar, Nepal, Pakistan, Philippines, Qatar, Russia, Saudi Arabia, Singapore, Syria, Taiwan, Thailand, Turkey, Cyprus, United Arab Emirates, Uzbekistan, Vietnam, and Yemen). *Blastocystis* spp. have been identified in humans, different kinds of animals, leafy vegetables, water, and ambient air using conventional microscopy, in vitro cultivation, and molecular methods.

## 3. *Blastocystis* spp. Infection in Humans

Investigations on human *Blastocystis* spp. infections were on children, high school and college students, hospital patients/patients referred to medical laboratories for tests, patients with gastrointestinal disorder (GID) and other conditions, immunocompromised individuals, different categories of workers, and apparently healthy and general populations from urban and rural settings alike. Irrespective of these human host groups, *Blastocystis* spp. were the common organisms detected in studies describing gastrointestinal tract organisms in humans; and *Blastocystis* spp. ST1, ST2 and ST3 were the most frequently isolated.

The presence of *Blastocystis* spp. has been reported in infants, kindergarten, and school-aged children in Asia in the past ten years (Table 1). However, the participants were either asymptomatic or their clinical conditions were not available. The majority of the studies were from Iran, Thailand, Malaysia, Turkey, and Indonesia; prevalence rates reported ranged from 1.2% to 83.7%. Only about 24% of these studies reported on *Blastocystis* spp. subtypes. Subtypes identified were ST1, ST2, ST3, ST4, ST5, ST6 and ST7.

Prevalence and subtypes of *Blastocystis* spp. in immunocompromised individuals in Asia are summarized in Table 2. This category comprised mostly cancer, HIV/AIDS, and pulmonary tuberculosis patients. Reported prevalence rates were generally not above 30% except 54.8% in immunocompromised children with diarrhea in Indonesia, and 42.2% and 53.6% prevalence in HIV/AIDS cases and pulmonary tuberculosis patients respectively in Uzbekistan. *Blastocystis* spp. subtypes 1, 2, 3, 4, 5, and 7 were identified.

It is noted that hematologic and non-hematologic (cranial) cancers with *Blastocystis* spp. infections are most commonly reported in children [82,83,84,85]. Whereas, colorectal, stomach, esophagus and non-gastrointestinal cancer such as lung, liver, breast, ovarian, hematologic and other cancers were detected in adults. Among the 10 studies focused on cancer patients, six studies clearly stated that cancer patients were receiving chemotherapy treatment [82,83,84,85,87,88]. One study recruited cancer patients who have not received any chemotherapy [89]. While the remaining two were classified as follow-up cases [90] and in- or out-patient cases [91], respectively. It is noted that the highest prevalence of *Blastocystis* spp. infection in cancer patients is detected in those who have not received chemotherapy [89] as compared to the other six studies. This could be due to the existing immunocompromised condition of the cancer patients that allowed an opportunistic infection to occur.

Patients with different gastrointestinal complaints and disorders such as constipation, abdominal pain, diarrhea, irritable bowel syndrome (IBS) and inflammatory bowel disease (IBD) have been examined for *Blastocystis* spp. infection with positive results recorded as shown in Table 3. The prevalence rate was as low as 0.5%, with the highest being 67.1% and all isolates belonged to *Blastocystis* spp. subtypes 1, 2, 3, 4, 5, 6, and 7.

The occurrence of *Blastocystis* spp. in mental rehabilitation centers was documented by several authors from Iran only (Table 4). Prevalence ranged from 4% to 55.2%; and out of all nine of these studies, only one reported the use of molecular methods wherein ST1, ST3 and ST9 were identified.

Studies on the status of *Blastocystis* spp. infection in hospital in- and out-patients are shown in Table 5. The diseases/illnesses of these patients were, however, not stated in the reports. Nonetheless, they did not show any gastrointestinal-related symptoms and volunteered as healthy participants in the gastrointestinal studies. As a result of their involvement, though asymptomatic, they were detected positive for *Blastocystis* spp. infection. Infection rate as low as 0.02% was recorded in 23,278 Saudi Arabian patients, while all (100%) of 15 hospital patients without any gastrointestinal complaints were found positive for *Blastocystis* spp. Asides *Blastocystis* spp. subtypes 1, 2, and 3 which were the most commonly observed, STs 6 and 7 were also commonly identified while STs 4 and 5 were few.

Table 6 is a summary of *Blastocystis* spp. infection in students and working populations in Asia between 2010 and 2021.

Food handlers and immigrant workers were commonly screened in Iran and Qatar, respectively. In addition to *Blastocystis* spp. subtypes 1, 2, and 3; ST6 was isolated from chicken slaughterhouse staff in Lebanon [199], and ST5 in pig handlers in Thailand [214].

The majority of the studies on *Blastocystis* spp. infections in humans in Asia within 2010 and 2021 were on general populations of apparently healthy status; such participants comprised urban dwellers, rural dwellers, and healthy control for immunocompromised persons. As depicted in Table 7, low prevalence rates of less than 5% and rates as high as 50% were reported from the different countries where these studies were undertaken, and various techniques were used for the detection of this protist. *Blastocystis* spp. subtypes reported were STs1-7 and ST10, whose only record was from Lebanon.

The presence of *Blastocystis* spp. in various other human categories that do not quite fit into those discussed above is summarized in Table 8.

## 4. *Blastocystis* spp. Infection in Animals

In Asia, *Blastocystis* spp. infection have been documented in hoofed mammals (Table 9), carnivores (Table 10), non-human primates (NHPs) (Table 11), birds (Table 12), rodents (Table 13), reptiles (Table 14), insects and some other mammalian groups (Table 15).

The prevalence of *Blastocystis* spp., reported in the last ten years, varied widely among the ungulates. Infection was mostly reported in livestock animals such as cattle, goats, sheep and pigs. *Blastocystis* spp. ST10 and ST14 were the most frequently isolated from deer, alpacas, cattle, yaks, sheep and goats, while ST1 and ST5 were the most common in pigs.

*Blastocystis* spp. has been isolated from carnivores, both domestic and wild, in Asia. Prevalence ranged from 0.6% to 100%, with STs 1–8 and ST10 being identified. NHPs have been commonly described to harbor *Blastocystis* spp., with a reported prevalence reaching a 100%. Genetic analyses have recognized ST1, ST2, and ST3 as being the most common in this group of mammals. Interestingly, *Blastocystis* spp. ST9 was isolated from ring-tailed lemur from China [250].

*Blastocystis* spp. infections in birds have been reported. Prevalence varied widely, however, subtype identification revealed ST6, S7, ST8 as the most frequently isolated. The isolation of *Blastocystis* spp. ST9 in chicken in Malaysia [15] is peculiar. Diverse genera of rodents have been found as hosts to *Blastocystis* spp. Although STs 1, 3, 5, 7 and 13 have been reported, ST4 and ST17 were the most frequently identified.

A few studies have reported on the infection of reptiles with *Blastocystis* spp. with the highest sample size being 19. Prevalence ranged from 26.3% to 100%, no subtype has yet been mentioned. Although studies are still few, cockroaches have been found as hosts to *Blastocystis* spp. Two out of six studies have described infection to the subtype level, ST2 was identified in China [418] while ST3 was identified in Malaysia [438].

Other animals found as hosts to *Blastocystis* spp. are the gray kangaroo, red-necked wallaby, sugar glider, rabbit, and hedgehog.

## 5. *Blastocystis* spp. in Food and Environmental Sources

In the past decade, the presence of *Blastocystis* spp. has been reported in tap water, river water, seawater, wells, fishponds, wastewater, food and even ambient air in Asia. The prevalence rate ranged from 2.1% to 100% in the various water sources, and 2.8% to 10.2% in leafy vegetables (Table 16). The only study on *Blastocystis* spp. in ambient air reported a prevalence of 1.4%. *Blastocystis* spp. subtype identification is only available for water sources. STs 1, 2, 3, 4, 6, 8, 10 have so far been recorded from water samples; and although the prevalence of ST3 was highest, ST1 was the most widespread subtype.

## 6. Distribution of *Blastocystis* spp. by Country

From 2010 till now, the identification of *Blastocystis* spp. has been described for a total of 31 Asian countries. Out of these 31, genetic characterization and *Blastocystis* spp. subtype identification was available for 22 countries. Figure 1 reveals the distribution of the subtypes of *Blastocystis* spp. in these countries with a glimpse of subtypes shared by humans, animals, and water sources. *Blastocystis* spp. ST1 was the most widespread subtype, found in all of the 22 countries.

## 7. Discussion

*Blastocystis* spp. have been reported in over 50% of the countries in the continent of Asia. Although the most documented hosts to infection were humans and several animal species, this organism has also been detected in water sources, vegetables, and ambient air.

Variation of prevalence rates was seen within and between the various human host categories. Although authors have described both significant and insignificant differences between *Blastocystis* spp. infection in patients with and without known disease conditions, this variation could be a result of the methods employed in the detection of *Blastocystis* spp. *Blastocystis* spp. STs 1–7 have been identified in humans in Asia. ST1, ST2, ST3 and ST4 were more widespread and more frequently isolated than ST5, ST6 and ST7. This finding is in agreement with studies from other parts of the world [2,13,452,453].

The isolation of *Blastocystis* spp. STs 1–14, and ST18–22 (novel subtypes) were reported in animal hosts. ST1, ST2, ST3, ST4, ST5, ST6 and ST7 were found common to humans and animals. ST9 was observed in ring-tailed lemurs and chickens in China [250] and Malaysia [15] respectively; however, no article included in this review reported on the identification of ST9 in humans in these countries. The characteristic presence of ST5 in pigs, ST10 and ST14 in goats, sheep and cattle, and ST6 and ST7 in chickens underscore suggestions that these STs are specific to the respective animal hosts. Also, reports of isolation of ST5 in pig handlers [214] and ST6 in chicken slaughterhouse staff [199] are pointers to possible zoonotic transmission.

Where stated, cysts were the *Blastocystis* spp. forms observed in vegetables and water samples. The presence of cysts in the life cycle of *Blastocystis* spp. enable their existence outside of human and animal hosts; also, the chloroform-resistant nature of these cysts probably explains the presence of *Blastocystis* spp. even in treated water.

## 8. Conclusions

The growing interest in the study of *Blastocystis* spp. as an area of research is very obvious and fundamental to unraveling the much that is hitherto unknown of the epidemiology, biology and pathogenicity of this protist. *Blastocystis* spp. have been isolated from biotic and abiotic sources in Asia. Considering that humans and animals are in constant interactions with their environment, epidemiological studies of *Blastocystis* spp. from an ecological perspective are essential. In essence, continuous surveillance of human and animal hosts alongside their food and water sources and other possible sources of infection such as soil across different geographical locations and climatic conditions is needed. The use of molecular detection methods in epidemiological studies are recommended to provide information on *Blastocystis* spp. STs in as many regions as possible. Incorporating the One Health (OH) method into epidemiological studies will equip researchers and other stakeholders with information on the possible influence of ecosystems on *Blastocystis* spp., it will further elucidate transmission routes and provide clues required to break the transmission of this protist successfully. Morphological studies of *Blastocystis* spp. in various host species and environmental sources are insufficient but essential; electron microscopy could help to accentuate structural details of isolates from various hosts and the differences or similarities between them, and contribute to the understanding of a proper, more detailed *Blastocystis* spp. lifecycle.

## Figures and Tables

**Figure 1 biology-10-00990-f001:**
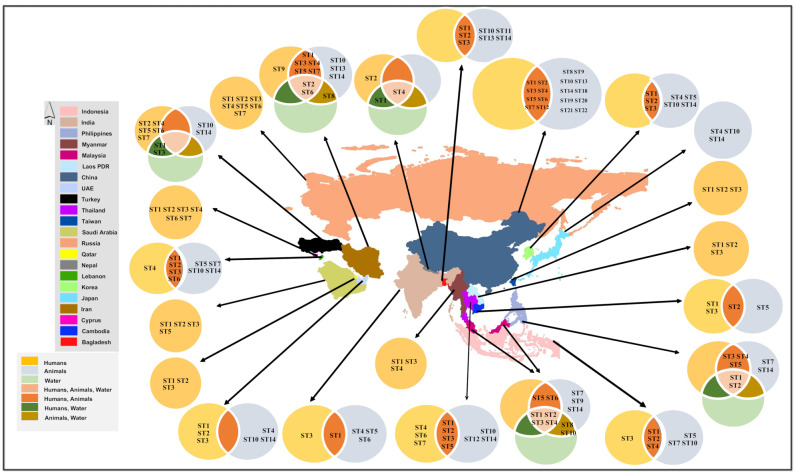
Distribution of *Blastocystis* spp. subtypes in Asia (2010–2021).

**Table 1 biology-10-00990-t001:** Prevalence and subtype distribution of *Blastocystis* spp. in children in Asia (2010–2021).

Country	No. of Samples Examined	No. of Positive Samples (%)	Subtypes (STs) Identified	Method(s)	References
Bangladesh	5679	795 (14.0)	NA	CM, 1VC	Barua et al. [34]
Israel	45,978	5422 (11.8)	NA	CM, IVC	Ben-Shimol et al. [35]
China	170	1 (0.5)	NA	MOL	Zhang et al. [36]
China	609	87 (14.3)	ST1, ST2, ST3	MOL	Qi et al. [37]
China	466	71 (15.2)	ST1, ST3, ST6, ST7	IVC, MOL	Ning et al. [38]
Cambodia	308	15 (4.9)	NA	CM	Liao et al. [39]
India	195	32 (16.4)	NA	CN	Rayan et al. [40]
Indonesia	492	147 (29.9)	ST1, ST2, ST3	IVC, MOL	Yoshikawa et al. [41]
Indonesia	99	33 (33.3)	ST1, ST2, ST3	MOL	Zulfa et al. [42]
Indonesia	141	58 (41.1)	ST1, ST3, ST4	IVC, MOL	Sari et al. [43]
Indonesia	219	15 (6.8)	NA	CM	Subahar et al. [44]
Indonesia	157	44 (28.0)	NA	CM	Sari et al. [45]
Iran	124,366	3986 (3.2)	NA	CM	Ashtiani et al. [46]
Iran	864	36 (4.1)	ST1, ST2, ST3	CM, IVC, MOL	Niaraki et al. [47]
Iran	366	11 (3.1)	NA	CM	Mahmoudvand et al. [48]
Iran	650	37 (5.7)	NA	CM	Abdi et al. [49]
Iran	1100	149 (13.5)	NA	CM	Daryani et al. [50]
Iran	350	15 (4.3)	NA	CM	Hazrati Tappeh et al. [51]
Iran	854	26 (3.0)	NA	CM	Norouzi et al. [52]
Iran	200	35 (17.5)	NA	CM	Babakhani et al. [53]
Iran	400	85 (21.3)	NA	CM	Bahmani et al. [54]
Iran	306	9 (2.9)	NA	CM	Saki and Amraee [55]
Iran	1465	31 (2.1)	NA	CM	Turki et al. [56]
Iraq	107	4 (3.7)	NA	CM	Mahdi and Al-Saadoon [57]
Lebanon	249	157 (63.0)	ST1, ST2, ST3	MOL	Osman et al. [58]
Malaysia	71	66 (93.0)	NA	CM	Abd. Ghani and Yusof [59]
Malaysia	300	77 (25.7)	NA	IVC	Abdulsalam et al. [60]
Malaysia	307	65 (21.2)	NA	CM	Al-Harazi et al. [61]
Malaysia	342	4 (1.2)	NA	CM	Sinniah et al. [62]
Malaysia	1760	186 (10.6)	ST1, ST2, ST3, ST4, ST5	IVC, MOL	Nithyamathi et al. [63]
Malaysia	116	2 (1.7)	NA	CM	Tang and Kamel [64]
Malaysia	92	77 (83.7)	NA	CM	Adli et al. [65]
Nepal	342	4 (1.2)	NA	CM	Mukhiya et al. [66]
Saudi Arabia	1289	11 (0.9)	NA	CM	Al-Mohammed et al. [67]
Saudi Arabia	581	10 (1.7)	NA	CM	Bakarman et al. [68]
Thailand	203	9 (4.4)	NA	CM	Suntaravitun and Dokmaikaw [69]
Thailand	1909	626 (32.8)	NA	CM	Sanprasert et al. [70]
Thailand	370	118 (31.9)	ST1, ST2, ST6	MOL	Thathaisong et al. [71]
Thailand	233	29 (12.5)	ST1, ST2, ST3	IVC, MOL	Pipatsatitpong et al. [72]
Thailand	299	10 (3.3)	NA	CM	Punsawad et al. [73]
Thailand	263	46 (17.5)	NA	CM, IVC	Assavapongpaiboon et al. [74]
Thailand	331	44 (13.3)	ST1, ST3	IVC, MOL	Boondit et al. [75]
Thailand	202	3 (1.5)	NA	CM	Kitvatanachai and Rhongbutsri [76]
Thailand	274	37 (13.5)	NA	CM	Popruk et al. [77]
Turkey	195	28 (14.4)	NA	CM	Güdücüoğlu et al. [78]
Turkey	328	77 (23.5)	NA	CM	Hamamci et al. [79]
Turkey	468	35 (7.4)	ST1, ST2, ST3, ST7	CM, IVC, MOL	Sankur et al. [80]
Turkey	1181	7 (0.6)	NA	CM	Calik et al. [81]
Turkey	219	97 (44.3)	ST1, ST2, ST3, ST4	MOL	Dogan et al. [82]

CM—Conventional microscopy, IVC—In vitro cultivation, MOL—Molecular technique, NA—Not applicable.

**Table 2 biology-10-00990-t002:** Prevalence and subtype distribution of *Blastocystis* spp. in immunocompromised individuals in Asia (2010–2021).

Host	Country	No. of Samples Examined	No. of Positive Samples (%)	Subtypes (STs) Identified	Method(s)	References
Cancer patients (children)	Iran	52	11 (21.2)	NA	CM	Salehi Kahish et al. [83]
Cancer patients (children)	Iran	200	24 (12.0)	ST1, ST2, ST3, ST7	MOL	Asghari et al. [84]
Cancer patients (children)	Iran	52	11 (21.2)	NA	CM	Salehi Kahyesh et al. [85]
Cancer patients (children)	Iran	89	5 (5.6)	NA	CM	Zabolinejad et al. [86]
Cancer patients	Iran	67	16 (23.9)	NA	CM, MOL	Mahmoudvand et al. [87]
Cancer patients	China	381	27 (7.1)	ST1, ST3	MOL	Zhang et al. [88]
Cancer patients	Malaysia	61	13 (21.3)	NA	IVC	Chandramathi et al. [89]
Cancer patients	Saudi Arabia	138	38 (27.5)	ST1, ST2, ST5	MOL	Mohamed et al. [90]
Cancer patients	Turkey	232	25 (10.8)	ST1, ST2, ST3	CM, IVC, MOL	Yersal et al. [91]
Cancer patients	Turkey	201	29 (14.4)	ST1, ST2, ST3	CM, MOL	Mulayim et al. [92]
HIV/AIDS cases	China	324	12 (3.7)	ST1, ST3, ST4, ST7, ST12	MOL	Teng et al. [93]
HIV/AIDS cases	China	720	154 (21.4)	NA	IVC	Tian et al. [94]
HIV/AIDS cases	China	302	49 (16.2)	NA	IVC	Tian et al. [95]
HIV/AIDS cases	China	79	11 (13.9)	NA	IVC	Tian et al. [96]
H IV/AIDS cases	China	398	27 (6.8)	NA	MOL	Zhang et al. [97]
HIV/AIDS cases	China	311	12 (3.9)	ST1, ST3, ST4, ST7	MOL	Zhang et al. [98]
HIV/AIDS cases	China	505	21 (4.2)	NA	MOL	Zhu-Hua et al. [99]
HIV/AIDS cases	India	452	13 (2.9)	NA	CM	Ramana et al. [100]
HIV/AIDS cases	India	200	14 (7.0)	NA	CM	Khalil et al. [101]
HIV/AIDS cases	Iran	31	7 (22.6)	NA	CM	Berenji et al. [102]
HIV/AIDS cases	Iran	60	10 (16.7)	NA	CM	Yosefi et al. [103]
HIV/AIDS cases	Iran	356	14 (3.9)	NA	CM	Agholi et al. [104]
HIV/AIDS cases	Iran	102	2 (1.9)	NA	CM	Masoumi-Asl et al. [105]
HIV/AIDS cases	Iran	73	2 (2.7)	NA	CM	Anvari-Tafti et al. [106]
HIV/AIDS cases	Iran	268	51 (19.0)	ST1, ST2, ST3, ST4	MOL	Piranshahi et al. [107]
HIV/AIDS cases	Laos	137	36 (26.3)	NA	CM	Paboriboune et al. [108]
HIV/AIDS cases	Nepal	146	9 (6.2)	NA	CM	Sherchan et al. 2012 [109]
HIV/AIDS cases	Nepal	112	1 (0.9)	NA	CM	Ghimire et al. [110]
HIV/AIDS cases	Turkey	65	7 (10.8)	NA	CM	Zorbozan et al. [111]
HIV/AIDS cases	Uzbekistan	500	211 (42.2)	NA		Davis et al. [112]
Tuberculosis	Iran	161	19 (11.8)	ST1, ST2, ST3	CM, MOL	Taghipour et al. [113]
Tuberculosis	Iran	161	19 (11.8)	NA	CM	Taghipour et al. [114]
Pulmonary tuberculosis	Uzbekistan	300	161 (53.6)	NA		Davis et al. [112]
Pulmonary tuberculosis	China	369	23 (6.2)	NA	CM, 1VC	Li et al. [115]
Pulmonary tuberculosis	China	369	23 (6.2)	NA	CM, 1VC	Li et al. [116]
pulmonary tuberculosis	Iran	50	9 (18.0)	NA	CM	Taghipour et al. [117]
Renal transplant recipients	Iran	150	7 (4.7)	NA	CM	Azami et al. [118]
Immunocompromised children with diarrhea	Indonesia	42	23 (54.8)	NA	IVC	Idris et al. [119]
Immunocompromised children with diarrhea	Turkey	62	6 (9.7)	NA	CM	Caner et al. [120]
Immunocompromised patients	Iran	265	11 (4.2)	NA	CM	Rasti et al. [121]
Immunocompromised patients	Iran	204	62 (30.4)	NA	CM	Izadi et al. [122]
Immunodeficient patients	Iran	190	32 (16.8)	NA	CM	Esteghamati et al. [123]
Immunosuppressive drugs recipient	Iran	494	49 (10.3)	NA	CM	Mirzaei et al. [124]
Immunocompromised patients	Saudi Arabia	136	7 (5.2)	NA	CM	Al-Megrin et al. [125]
Common variable immune deficiency (CVID) syndrome patients	Turkey	37	3 (8.1)	NA	CM	Uysal et al. [126]

CM—Conventional microscopy, IVC—In vitro cultivation, MOL—Molecular technique, NA—Not applicable.

**Table 3 biology-10-00990-t003:** Prevalence and subtype distribution of *Blastocystis* spp. in humans with gastrointestinal symptoms in Asia (2010–2021).

Host	Country	No. of Samples Examined	No. of Positive Samples (%)	Subtypes (STs) Identified	Method(s)	References
Children with diarrhea	China	850	26 (3.1)	NA	MOL	Zhang et al. [36]
Children with diarrhea	Indonesia	57	36 (63.1)	ST1, ST2, ST3, ST4	MOL	Zulfa et al. [42]
Children with diarrhea	Turkey	60	4 (6.7)	NA	CM	Maçin et al. [127]
Children with diarrhea	Iran	400	8 (2.0)	NA	CM	Asadi et al. [128]
Children with diarrhea	Qatar	580	27 (4.7)	NA	MOL	Boughattas et al. [129]
Children with diarrhea	Nepal	588	5 (0.9)	NA	CM	Dahal et al. [130]
Children with diarrhea	Iran	160	37 (23.1)	NA	CM	Khalili et al. [131]
Children with GID	Iran	500	81 (16.2)	NA	CM	Kiani et al. [132]
Children with GID	Thailand	82	13 (15.9)	NA	CM, IVC	Awae et al. [133]
Children with GID	Russia	1273	62 (4.9)	ST1, ST2, ST3, ST4, ST7	CM, MOL	Sigidaev et al. [134]
Children with GID	Turkey	84	18 (21.4)	ST1, ST3, ST4	MOL	Dogan et al. [82]
Patients with diarrhea	Indonesia	389	22 (5.7)	NA	CM	Oyofo et al. [135]
Patients with diarrhea	China	271	13 (4.8)	NA	MOL	Zhang et al. [136]
Patients with diarrhea	Korea	117	8 (6.8)	NA	MOL	Won et al. [137]
Patients with diarrhea	Iran	134	28 (20.9)	ST1, ST2, ST3	CM, MOL	Jalallou et al. [138]
Patients with diarrhea	Iran	2023	1357 (67.1)	NA	CM	Najafi et al. [139]
Patients with GID	Iran	1301	350 (26.9)	NA	CM	Kiani et al. [140]
Patients with GID	Iran	287	65 (22.7)	ST1, ST2, ST3, ST5	IVC, MOL	Moosavi et al. [141]
Patients with GID	Iran	23*	23	ST1	CM, MOL	Shahbazi et al. [142]
IBD patients	Iran	71	9 (12.7)	ST1, ST3	IVC, MOL	Mirjalali et al. [143]
Adolescents with IBS	Indonesia	137	50 (36.5)	ST1, ST2, ST3	MOL	Kesuma et al. [144]
IBS patients	India	150	50 (33.3)	ST1, ST3	CM, IVC, MOL	Das et al. [145]
IBS patients	Iran	100	15 (15.0)	NA	CM	Shafiei et al. [146]
IBS patients	Iran	122	24 (19.7)	ST1, ST3, ST4, ST5	MOL	Khademvatan et al. [147]
IBS patients	Iraq	78	38 (48.7)	NA	CM, IVC	Sayal et al. [148]
IBS patients	Thailand	66	11 (16.7)	NA	IVC	Surangsrirat et al. [149]
Patients with GID	Iraq	579	98 (16.9)	NA	CM, IVC	Merza et al. [150]
Patients with GID	Iraq	249	92 (36.9)	NA	CM	Mutlag et al. [151]
Patients with GID	Thailand	5	5 (100.0)	ST3, ST6, ST7	CM, IVC, MOL	Sanpool et al. [152]
Patients with diarrhea	Turkey	272	16 (5.9)	NA	CM, MOL	Koltas et al. [153]
Patients with GID	Turkey	490	89 (18.2)	NA	CM, IVC	Aykur et al. [154]
Patients with GID	Turkey	14,246	689 (4.8)	NA	CM	Usluca et al. [155]
Patients with GID	Turkey	2334	134 (5.7)	NA	CM	Cekin et al. [156]
Patients with GID	Iran	152	16 (10.5)	ST1, ST2, ST3	CM, IVC, MOL	Beiromvand et al. [157]
Patients with diarrhea	Singapore	193	1 (0.5)	NA	CM, MOL	Feurle et al. [158]
Patients with GID	Saudi Arabia	114	15 (13.2)	NA	CM	Hawash et al. [159]
Patients with GID	Turkey	5624	136 (2.4)	NA	CM	Alver et al. [160]
Patients with GID	Turkey	17756	778 (4.4)	NA	CM	İnceboz et al. [161]
Patients with GID	Iran	670	38 (5.7)	NA	IVC	Rostami Nejad et al. [162]
Patients with GID	Pakistan	339	59 (17.4)	NA	CM	Haider et al. [163]
Patients with GID	Turkey	29 *	29	ST1, ST2, ST3, ST4	CM, MOL	Sakalar et al. [164]

* Study was carried out on *Blastocystis* sp. positive hosts, CM—Conventional microscopy, IVC—In vitro cultivation, MOL—Molecular technique, NA—Not applicable, GID—Gastrointestinal disorder, IBD—Inflammatory bowel disease, IBS—Irritable bowel syndrome.

**Table 4 biology-10-00990-t004:** Prevalence and subtype distribution of *Blastocystis* spp. in mental rehabilitation centers in Asia (2010–2021).

Host	Country	No. of Samples Examined	No. of Positive Samples (%)	Subtypes (STs) Identified	Method(s)	References
Mentally disabled children	Iran	362	20 (5.5)	NA	CM	Sharif et al. [165]
Mentally disabled children and adults	Iran	225	9 (4.0)	NA	CM	Hazrati Tappeh et al. [166]
Psychiatric patients	Iran	65	15 (23.1)	NA	CM	Khalili et al. [167]
Mentally disabled individuals	Iran	173	29 (16.8)	NA	CM	Saeidinia et al. [168]
Mentally disabled individuals	Iran	133	12 (9.0)	NA	CM	Shokri et al. [169]
Mentally disabled individuals and elderly people	Iran	243	81 (33.3)	NA	CM	Rasti et al. [170]
Mentally disabled individuals	Iran	126	38 (30.2)	NA	CM	Mohammadi-Meskin et al. [171]
Mental retardation center personnel	Iran	37	12 (32.4)	NA	CM	Mohammadi-Meskin et al. [171]
Schizophrenic male patients	Iran	58	32 (55.2)	ST1, ST3, ST9	CM, MOL	Sheikh et al. [172]

CM—Conventional microscopy, MOL—Molecular technique, NA—Not applicable.

**Table 5 biology-10-00990-t005:** Prevalence and subtype distribution of *Blastocystis* spp. in patients of health institutions in Asia (2010–2021) who volunteered in gastrointestinal studies.

Country	No. of Samples Examined	No. of Positive Samples (%)	Subtypes (STs) Identified	Method(s)	References
China	126	3 (2.4)	ST5	MOL	Zhu et al. [173]
China	198	21 (10.6)	ST1, ST3, ST6, ST7	MOL	Kang et al. [174]
Iran	670	23 (3.4)	NA	IVC	Rostami Nejad et al. [162]
Iran	1232	154 (12.6)	NA	CM	Abdipour et al. [175]
Iran	1383	239 (17.3)	ST1, ST2, ST3	CM, MOL	Bahrami et al. [176]
Iran	984	13 (1.3)	NA	CM	Gholipoor et al. [177]
Iran	417	39 (9.4)	NA	CM	Viesy et al. [178]
Iran	511	33 (6.5)	ST2, ST3, ST5	MOL	Badparva et al. [179]
Iran	420	60 (14.3)	ST2, ST3	CM, MOL	Shaker et al. [180]
Iran	802	39 (4.9)	ST1, ST2, ST3, ST7	MOL	Haghighi et al. [181]
Iran	420	60 (14.3)	NA	CM	Shaker et al. [182]
Iran	1120	65 (5.8)	NA	CM	Tork et al. [183]
Iran	4788	247 (5.2)	NA	CM	Asfaram et al. [184]
Iran	210	66 (31.4)	ST1, ST2, ST3, ST4, ST5, ST6	MOL	Bafghi et al. [185]
Iran	133	35 (26.3)	ST1, ST2, ST3, ST5	IVC, MOL	Moosavi et al. [141]
Iran	4427	407 (9.2)	NA		Karimazar et al. [186]
Iraq	300	146 (48.7)	NA	CM	Abdul Ridha and Faieq, [187]
Iran	618	146 (23.6)	ST1, ST2, ST3	CM, IVC, MOL	Salehi et al. [188]
Iran	481	69 (14.4)	ST1, ST2, ST3, ST4, ST5	MOL	Khademvatan et al. [189]
Iran	250	41 (16.4)	ST1, ST2, ST3	CM, IVC, MOL	Sardarian et al. [190]
Iran	200	63 (31.5)	NA	CM, IVC	Hamidi et al. [191]
Iran	5000	784 (1.6)	NA	CM	Javadi et al. [192]
Iran	864	68 (7.9)	ST1, ST2, ST3	CM, IVC, MOL	Delshad et al. [193]
Iran	566	10 (1.8)	NA	CM	Norouzi et al. [194]
Iran	100	13 (13.0)	ST1, ST2, ST6	CM, MOL	Sharifi et al. [195]
Iran	1878	152 (8.1)	ST1, ST2, ST3, ST7	CM, MOL	Salehi et al. [196]
Lebanon	40	23 (57.5)	ST1, ST2, ST3	MOL	Greige et al. [197]
Lebanon	220	42 (19.1)	ST1, ST2, ST3, ST4	CM, MOL	El Safadi et al. [198]
Lebanon	50	27 (54.0)	ST1, ST2, ST3	MOL	Greige et al. [199]
Saudi Arabia	23,278	5 (0.02)	NA	CM	Imam et al. [200]
Saudi Arabia	130	3 (2.3)	NA	CM	Hassen Amer et al. [201]
Saudi Arabia	1262	133 (10.5)	ST1, ST2, ST3	IVC, MOL	Mohamed et al. [202]
Thailand	14,325	199 (1.4)	NA	CM	Laodim et al. [203]
Thailand	562	56 (9.9)	ST1, ST3, ST6, ST7	IVC, MOL	Jantermtor et al. [204]
Thailand	15	15 (100.0)	ST1, ST3, ST6, ST7	CM, IVC, MOL	Sanpool et al. [152]
Turkey	192	6 (3.1)	NA	CM	Cekin et al. [156]
Turkey	20,948	13,245 (63.2)	NA	CM	Polat et al. [205]
Turkey	50,185	275 (0.5)	NA	CM	Beyhan et al. [206]
Turkey	4030	476 (11.1)	ST1, ST2, ST3	CM, MOL	Sarzhanov et al. [207]
Turkey	6757	160 (2.4)	NA	CM	Selek et al. [208]

CM—Conventional microscopy, IVC—In vitro cultivation, MOL—Molecular technique, NA—Not applicable.

**Table 6 biology-10-00990-t006:** Prevalence and subtype distribution of *Blastocystis* spp. in students and working populations in Asia (2010–2021).

Host	Country/Region	No. of Samples Examined	No. of Positive Samples (%)	Subtypes (STs) Identified	Method(s)	References
Adolescents	Indonesia	70	20 (28.6)	ST1, ST3	MOL	Kesuma et al. [144]
High school students foreign	Turkey	192	63 (32.8)	NA		Yaman et al. [209]
College students	China	53 *	53	ST1, ST3, ST4, ST6, ST7	IVC, MOL	Zhan et al. [210]
College students of practical parasitology courses	Iran	175	9 (5.1)	NA	CM	Fallahi et al. [211]
Students who did not take any practical parasitology courses	Iran	135	5 (3.7)	NA	CM	Fallahi et al. [211]
University students	Thailand	1025	416 (40.6)	ST1, ST2, ST3	CM, IVC, MOL	Srichaipon et al. [212]
Working children	Iran	175	57 (32.6)	NA	CM	Salemi et al. [213]
Caregivers in a childcare center	Thailand	25	6 (24.0)	ST1, ST2, ST3	IVC, MOL	Pipatsatitpong et al. [72]
Cattle breeders	Lebanon	40	21 (52.5)	ST1, ST2, ST3	MOL	Greige et al. [197]
Chicken slaughterhouse staff	Lebanon	50	28 (56.0)	ST1, ST2, ST3, ST6	MOL	Greige et al. [199]
Pig handler and individuals who lived near pig farms	Thailand	154	10 (6.5)	ST1, ST3, ST5	MOL	Pintong et al. [214]
Pig handler and individuals who lived near pig farms	Thailand	117	15 (12.8)	ST1, ST2, ST3	MOL	Udonsom et al. [215]
Food handlers	Iran	210	3 (1.4)	NA	CM	Kheirandish et al. [216]
Food handlers	Iran	1021	40 (3.9)	NA	CM	Motazedian et al. [217]
Food handlers	Iran	1041	29 (2.8)	NA	CM	Sharif et al. [218]
Food handlers	Iran	800	194 (24.3)	NA	CM	Heydari-Hengami et al. [219]
Food handlers	Iran	1018	7 (7.2)	NA	CM	Khodabakhsh Arbat et al. [220]
Food handlers	Iran	1530	44 (2.9)	NA	CM	Shahnazi et al. [221]
food handlers	Iran	816	18 (2.2)	NA	CM	Kheirandish et al. [222]
Food handlers	Jordan	901	6 (0.7)	NA	CM	Abdel-Dayem et al. [223]
Military personnel	Iraq	437	36 (8.2)	NA	CM	Downs et al. [224]
Gardeners	Thailand	253	23 (9.1)	NA	CM	Kitvatanachai and Rhongbutsri, [225]
Immigrant workers	Thailand	600	6 (1.0)	NA	CM	Sangwalee et al. [226]
Immigrant workers	Qatar	608	432 (71.1)	ST1, ST2, ST3	MOL	Abu-Madi et al. [227]
Immigrant workers	Qatar	735	479 (65.2)	NA	CM, MOL	Abu-Madi et al. [228]
Settled immigrant	Qatar	9208	398 (4.3)	NA	CM	Abu-Madi et al. [229]
Newly arrived immigrants	Qatar	2486	137 (5.5)	NA	MOL	Abu-Madi et al. [230]
Settled immigrants	Qatar	29,286	1010 (3.5)	NA	MOL	Abu-Madi et al. [231]
Resident workers	Qatar	772	39 (5.1)	NA	CM	Abu-Madi et al. [232]
Workers	Saudi Arabia	1238	245 (19.8)	NA	CM	Wakid [233]
New employees in a tertiary health care center	Saudi Arabia	2490	314 (12.6)	NA		Ahmed et al. [234]
Foreign laborers	Taiwan	7360	190 (2.6)	NA	CM	Hsieh et al. [235]
Foreigners	Taiwan	2875	33 (1.1)	NA	CM	Hsieh et al. [236]
Indonesian immigrant workers	Taiwan	128	28 (21.9)	ST1, ST2, ST3	CM, MOL	Chen et al. [237]
Sanitary and Non-sanitary Institutions’ workers	Turkey	2443	175 (7.2)	NA	CM	Karaman et al. [238]
Migrant workers	Malaysia	220	68 (30.9)	ST1, ST2, ST3	IVC, MOL	Sahimin et al. [239]

* Study was carried out on *Blastocystis* spp. positive hosts, CM—Conventional microscopy, IVC—In vitro cultivation, MOL—Molecular technique, NA—Not applicable.

**Table 7 biology-10-00990-t007:** Prevalence and subtype distribution of *Blastocystis* spp. in apparently healthy general populations in Asia (2010–2021).

Country	No. of Samples Examined	No. of Positive Samples (%)	Subtypes (STs) Identified	Method(s)	References
Bangladesh	140	51 (36.4)	NA	CM	Noor et al. [240]
Cambodia	218	40 (18.4)	NA	CM	Schär et al. [241]
Cambodia	210	116 (55.2)	ST1, ST2, ST3	MOL	Wang et al. [242]
China	497	215 (43.3)	NA	CM	He et al. [243]
China	5939	494 (8.3)	NA	MOL	Chen et al. [244]
China	26,886	2 (0.01)	NA	CM	Umar et al. [245]
China	1023	1 (0.1)	NA	CM	Jiang [246]
China	6710	19 (0.3)	NA	CM	Zhang et al. [247]
China	303	67 (22.1)	NA	IVC	Tian et al. [95]
China	294	64 (21.8)	NA	IVC	Tian et al. [96]
China	149	9 (6.0)	NA	MOL	Zhang et al. [136]
China	366	28 (7.6)	NA	CM, 1VC	Li et al. [115]
China	289	13 (4.5)	ST1, ST3, ST4	MOL	Gong et al. [248]
China	507	48 (9.5)	ST1, ST2, ST3, ST4	MOL	Deng et al. [249]
China	1118	390 (34.9)	ST2, ST5	MOL	Ma et al. [250]
Cyprus	230	64 (27.8)	ST1, ST2, ST3, ST4, ST6, ST7	MOL	Seyer et al. [251]
India	279	105 (37.6)	NA	MOL	Padukone et al. [252]
India	200	16 (8.0)	NA	CM	Khalil et al. [101]
India	100	15 (15.0)	ST1, ST3	CM, IVC, MOL	Das et al. [145]
India	23	13 (56.5)	NA	MOL	Lappan et al. [253]
Indonesia	646	15 (2.3)	NA	CM	Wiria et al. [254]
Indonesia	54	5 (9.3)	NA	IVC	Yulfi et al. [255]
Indonesia	424	146 (34.4)	NA	CM	Sungkar et al. [256]
Indonesia	53	9 (17.0)	NA	CM	Hayashi et al. [257]
Iran	5073	368 (7.3)	NA	CM	Turgay et al. [258]
Iran	399	16 (4.0)	NA	CM	Mahmoudi et al. [259]
Iran	130	40 (30.1)	ST1, ST2, ST3	CM, IVC, MOL	Beiromvand et al. [157]
Iran	20	3 (15.0)	NA	CM	Berenji et al. [102]
Iran	166	35 (21.1)	ST1, ST2, ST3	IVC, MOL	Mirjalali et al. [143]
Iran	181	17 (9.4)	NA	CM	Taghipour et al. [114]
Iran	225	5 (2.2)	NA	CM	Azami et al. [118]
Iran	166	35 (21.1)	ST1, ST2, ST3	CM, MOL	Jalallou et al. [138]
Iran	147	0 (0.0)	NA	CM	Anvari-Tafti et al. [106]
Iran	122	21 (17.2)	ST1, ST3, ST4, ST5	MOL	Khademvatan et al. [147]
Iran	100	6 (6.0)	NA	CM	Shafiei et al. [146]
Iran	67	6 (9.0)	NA	CM, MOL	Mahmoudvand et al. [87]
Iran	250	41 (16.4)	ST1, ST2, ST3	CM, IVC, MOL	Sardarian et al. [190]
Iran	1410	47 (3.3)	ST3, ST4, ST5, ST7	CM, MOL	Khoshnood et al. [260]
Iran	655	180 (27.5)	NA	CM	Pestehchian et al. [261]
Iran	5743	54 (0.9)	NA	CM	Sadeghi et al. [262]
Iran	5739	30 (0.5)	NA	CM	Sadeghi and Borji [263]
Iran	2838	139 (5.0)	NA	CM	Badparva et al. 2014 [264]
Iran	1060	145 (13.7)	NA	CM	Mahni et al. [265]
Iran	880	55 (6.3)	NA	CM	Tork et al. [266]
Iran	652	48 (7.4)	NA	CM	Jafari et al. [267]
Iran	561	159 (28.4)	NA	CM	Hemmati et al. [268]
Iran	554	93 (16.8)	NA	CM, IVC	Riabi et al. [269]
Iran	345	85 (24.6)	ST1, ST2, ST3	CM, IVC, MOL	Mardani Kataki et al. [270]
Iran	861	114 (13.2)	NA	CM	Abbaszadeh Afshar et al. [271]
Iran	732	63 (6.3)	NA	CM	Sobati [272]
Iran	184	45 (24.5)	ST1, ST2, ST3	MOL	Shirvani et al.[273]
Iran	283	20 (7.1)	NA	CM	Barati et al.[274]
Iran	2838	129 (4.5)	NA	CM	Badparva et al. [264]
Iran	565	144 (25.5)	NA	CM	Bairami Kuzehkanani et al. [275]
Iran	1025	182 (17.8)	NA	CM	Sarkari et al. [276]
Iran	1500	13 (0.9)	NA	CM	Sharifdini et al. [277]
Iran	4788	277 (5.8)	NA	CM	Pagheh et al. [278]
Iran	1008	46 (4.6)	NA	CM	Beiromvand et al. [279]
Iran	2280	81 (3.6)	NA	CM	Taherkhani et al. [280]
Iraq	78	1 (1.3)	NA	CM, IVC	Sayal et al. [148]
Korea	324	29 (9.0)	ST1, ST2, ST3	MOL	Kim et al. [281]
Laos	669	91 (13.6)	NA	CM	Sayasone et al. [282]
Laos	305	45 (14.8)	NA	CM	Ribas et al. [283]
Laos	60	32 (51.7)	ST1, ST2, ST3, ST7	CM, IVC, MOL	Sanpool et al. [284]
Lebanon	7477	178 (2.3)	NA	CM	Araj et al. [285]
Lebanon	306	195 (63.7)	ST1, ST2, ST3, ST10	MOL	Khaled et al. [286]
Malaysia	77	4 (5.2)	NA	CM	Sinniah et al. [287]
Malaysia	500	102 (20.4)	NA	CM	Anuar et al. [17]
Malaysia	243	45 (18.5)	ST1, ST2, ST3	MOL	Mohammad et al. [288]
Malaysia	466	191 (41.0)	NA	CM, IVC, MOL	Noradilah et al. [289]
Malaysia	253	103 (40.7)	NA	IVC	Mohammad et al. [290]
Malaysia	473	191 (40.4)	ST1, ST2, ST3, ST4	MOL	Noradilah et al. [291]
Malaysia	466	191 (41.0)	NA	CM, IVC, MOL	Noradilah et al. [292]
Malaysia	253	45 (17.8)	ST1, ST2, ST3	MOL	Mohammad et al. [293]
Malaysia	416	18 (4.3)	NA	CM	Muslim et al. [294]
Myanmar	172	16 (9.3)	ST1, ST3, ST4	MOL	Gong et al. [248]
Nepal	241	63 (26.1)	ST1, ST2, ST4	IVC, MOL	Lee et al. [295]
Philippines	110	36 (32.7)	NA	IVC	Santos and Rivera [296]
Philippines	1271	166 (13.0)	ST1, ST2, ST3, ST4, ST5	IVC, MOL	Belleza et al. [297]
Philippines	35	29 (82.9)	ST1, ST3, ST4	MOL	Adao et al. [298]
Philippines	1271	165 (13.0)	NA	IVC	Belleza et al. [299]
Philippines	412	242 (58.7)	NA	MOL	Weerakoon et al. [300]
Saudi Arabia	140	96 (68.6)	NA	CM	AlDahhasi et al. [301]
Saudi Arabia	80	12 (15.0)	ST1, ST2, ST5	MOL	Mohamed et al. [90]
Saudi Arabia	50	4 (8.0)	NA	CM	Hawash et al. [302]
Saudi Arabia	90	2 (2.2)	NA	CM	Hawash et al. [159]
Saudi Arabia	795	131 (16.5)	NA	CM	Alqumber [303]
Saudi Arabia	795	209 (26.3)	NA	CM	Alqumber [303]
Thailand	249	1 (0.4)	NA	CM	Kaewpitoon et al. [304]
Thailand	60	6 (10.0)	NA	IVC	Surangsrirat et al. [149]
Thailand	475	58 (12.2)	NA	CM, IVC	Kaewjai et al. [305]
Thailand	230	25 (10.8)	ST1, ST3, ST4	MOL	Popruk et al. [306]
Thailand	1047	29 (2.8)	NA	CM	Prommi et al. [307]
Thailand	178	41 (23.0)	ST1, ST2 ST3, ST4, ST6, ST7	MOL	Yowang et al. [308]
Thailand	324	13 (4.0)	NA	CM	Punsawad et al. [309]
Thailand	220	13 (5.9)	ST2, ST3, ST6	MOL	Palasuwan et al. [310]
Thailand	247	2 (0.8)	NA	CM	Kitvatanachai et al. [311]
Thailand	253	4 (1.6)	NA	CM	Boonjaraspinyo et al. [312]
Thailand	224	1 (0.4)	NA	CM	Suntaravitun and Dokmaikaw [313]
Thailand	733	57 (7.8)	NA	IVC	Wongthamarin et al. [314]
Thailand	207	77 (37.2)	ST1, ST2, ST3, ST4	MOL	Popruk et al. [315]
Turkey	30	4 (13.0)	NA	CM, MOL	Karasartova et al. [316]
Turkey	150	16 (10.7)	NA	CM	Karadag et al. [317]
Turkey	105	30 (28.6)	NA	IVC	Dogruman-Al et al. [318]
Turkey	27,664	581 (2.1)	NA	CM	Koksal et al. [319]
Turkey	176	30 (17.0)	NA	CM	Alver et al. [160]
Turkey	16,445	2602 (15.8)	NA	CM	Çetinkaya et al. [320]
Turkey	17,711	1353 (7.6)	NA	CM	Düzyol et al. [321]
Turkey	251	54 (21.5)	NA	CM	Kurt et al. [322]
Turkey	6267	968 (15.4)	NA	CM	Yılmaz et al. [323]
Turkey	87,100	640 (0.7)	NA		Gülmez et al. [324]
Turkey	111,889	306 (0.3)	NA	CM	Kirkoyun Uysal et al. [325]
Turkey	7353	1884 (63.6)	NA	CM	Öncel [326]
Turkey	200	93 (46.5)	ST1, ST2, ST3, ST7	MOL	Malatyalı et al. [327]
Turkey	69,633	18,460 (26.5)	NA	CM	Taş Cengiz et al. [328]
Turkey	104	10 (9.6)	ST1, ST2, ST3, ST6	MOL	Gulhan et al. [329]
Turkey	56	28 (50.0)	ST1, ST2, ST3, ST4, ST5, ST6, ST7	MOL	Koltas and Eroglu [330]
United Arab Emirates	133	59 (44.4)	ST1, ST2, ST3	MOL	AbuOdeh et al. [331]
Uzbekistan	300	31 (10.3)	NA	CM	Toychiev et al. [332]
Uzbekistan	550	99 (18.0)	NA	CM	Davis et al. [112]

CM—Conventional microscopy, IVC—In vitro cultivation, MOL—Molecular technique, NA—Not applicable.

**Table 8 biology-10-00990-t008:** Prevalence and subtype distribution of *Blastocystis* spp. in various human categories in Asia (2010–2021).

Host	Country	No. of Samples Examined	No. of Positive Samples (%)	Subtypes (STs) Identified	Method(s)	References
Acute appendicitis patients	Turkey	136	8 (5.9)	NA	CM	Hatipoğlu et al. [333]
Adult male prison inmates	Malaysia	294	43 (14.6)	ST1, ST3, ST6	CM, IVC, MOL	Angal et al. [334]
Adults with intestinal parasitic infection	Malaysia	35	17 (48.0)	NA	IVC	Chandramathi et al. [335]
Asymptomatic *Blastocystis* positive patients	Iran	25 *	25	ST1, ST2, ST3, ST7	MOL	Rezaei Riabi et al. [336]
Asymptomatic *Blastocystis* positive patients	Iran	34 *	34	ST2, ST3	CM, MOL	Shahbazi et al. [142]
Chronic spontaneous urticaria (adults)	Turkey	38	7 (18.4)	NA	CM	Vezir et al. [337]
Chronic spontaneous urticaria (children)	Turkey	76	13 (17.1)	NA	CM	Vezir et al. [337]
Urticarial patients	Turkey	133	16 (12.0)	ST1, ST2, ST3	CM, MOL	Aydin et al. [338]
Cirrhotic patients	Turkey	37	8 (21.6)	ST1, ST2, ST3	MOL	Yildiz et al. [339]
Diarrheic and non-diarrheic patients	Iran	400	58 (14.5)	ST1, ST2, ST3	IVC, MOL	Alinaghizade et al. [340]
Dengue patients	Malaysia	89	21 (23.6)	ST1, ST3, ST4, ST6	IVC, MOL	Thergarajan et al. [341]
Dialysis patients	Turkey	142	34 (23.9)	NA	CM	Karadag et al. [317]
*Giardia intestinalis* positive patients	India	258	21 (8.1)	NA	CM	Roy et al. [342]
Hemodialysis patients	Iran	88	8 (9.0)	NA	CM	Barazesh et al. [343]
Immunocompromised and control	Iran	641	57 (8.9)	NA	CM	Mahmoudi et al. [259]
Orphanage (orphansand childcare workers)	Thailand	343	94 (27.4)	NA	CM, IVC	Pipatsatitpong et al. [344]
Patients suspected to have intestinalparasites	Turkey	918	38 (4.2)	NA	CM	Koltas et al. [345]
Patients with chronic renal failure	Saudi Arabia	50	8 (16.0)	NA	CM	Hawash et al. [302]
Patients with chronic viral Hepatitis C	Russia	327	108 (33.0)	ST3, ST5, ST6	CM, MOL	Sigidaev et al. [134]
Patients with Erythema Nodosum	Turkey	81	2 (2.5)	NA		Ozbagcivan et al. [346]
Patients with gastrointestinal and/or dermatologic symptoms	Turkey	37,108	2537 (6.8)	NA	CM	Tunalı et al. [347]
Patients with intestinal protozoan infections	Iran	75	5 (6.7)	NA	CM	Jafari et al. [348]
Patients with systemic lupus erythematosus (SLE)	Malaysia	187	1 (0.5)	NA	not stated	Teh et al. 2018 [349]
Post-traumatic splenectomized patients	Turkey	30	12 (40.0)	ST1, ST3	CM, MOL	Karasartova et al. [316]
Pregnant women	Turkey	100	14 (14.0)	ST1, ST2, ST3	CM, IVC, MOL	Malatyalı et al. [350]
Symptomatic *Blastocystis* positive patients	Iran	30 *	30	ST1, ST2, ST3, ST6	MOL	Rezaei Riabi et al. [336]
Ulcerative colitis patients with refractory symptoms	China	49	6 (12.2)	NA	CM	Tai et al. [351]
Ulcerative colitis patients responsive to treatment	China	73	1 (1.4)	NA	CM	Tai et al. [351]
Visceral Leishmaniasis cases	India	23	14 (60.9)	NA	MOL	Lappan et al. [253]

* Study was carried out on *Blastocystis* spp. positive hosts, CM—Conventional microscopy, IVC—In vitro cultivation, MOL—Molecular technique, NA—Not applicable.

**Table 9 biology-10-00990-t009:** Prevalence and subtype distribution of *Blastocystis* spp. in ungulates in Asia (2010–2021).

Host	Country	No. of Samples Examined	No. of Positive Samples (%)	Subtypes (STs) Identified	Method(s)	References
Artiodactyla						
Alpaca	China	14	12 (85.7)	ST10, ST14, ST18	MOL	Zhao et al. [352]
Alpaca	China	27	4 (14.8)	ST10, ST14	MOL	Li et al. [353]
Alpaca	China	366	87 (23.8)	ST5, ST10, ST14	MOL	Ma et al. [354]
Alpaca	China	11	4 (36.4)	ST10, ST14	MOL	Deng et al. [3]
Blesbuck	China	2	1 (50.0)	ST5	MOL	Li et al. [353]
Buffalo	India	1	1 (100.0)	NA	CM	Sreekumar et al. [355]
Buffalo	Nepal	19	4 (21.1)	ST4	IVC, MOL	Lee et al. [18]
Bushbuck	China	18	8 (61.5)	ST10, ST14	MOL	Zhao et al. [352]
Camel	China	10	5 (50.0)	ST1, ST10	MOL	Zhao et al. [352]
Camel	China	40	14 (35.0)	ST2, ST10, ST14	MOL	Zhang et al. [14]
Cattle	Lebanon	254	161 (63.4)	ST1, ST2, ST3, ST5, ST7, ST10, ST14	MOL	Greige et al. [197]
Cattle	Malaysia	29	10 (34.5)	NA	IVC	Hemalatha et al. [356]
Cattle	Malaysia	3	1 (33.3)	ST10	MOL	Mohammad et al. [288]
Cattle	Malaysia	110	6 (5.4)	NA	IVC	Abd Razak et al. [357]
Cattle	Malaysia	80	35 (43.8)	ST1, ST3, ST4, ST5, ST10, ST14	MOL	Kamaruddin et al. [358]
Cattle	Nepal	6	1 (16.7)	Unknown	IVC, MOL	Lee et al. [18]
Cattle	Thailand	42	21 (50.0)	ST10, ST12	MOL	Udonsom et al. [215]
Cattle	Turkey	80	9 (11.3)	ST10, ST14	MOL	Aynur et al. [359]
Cattle	Indonesia	500	72 (14.4)	NA	CM	Hastutiek et al. [360]
Cattle	Indonesia	100	100 (100.0)	NA	CM	Susana et al. [361]
Cattle	Indonesia	108	108 (100.0)	ST10	CM, IVC, MOL	Suwanti et al. [362]
Cattle	Iran	198	19 (9.6)	ST3, ST5, ST6	MOL	Badparva et al. [363]
Cattle	Iran	75	25 (33.3)	ST5, ST10	CM, MOL	Sharifi et al. [195]
Cattle	Iran	40	14 (35.0)	ST3, ST10, ST14	CM, MOL	Rostami et al. [364]
Cattle	Japan	133	72 (54.1)	ST10, ST14	MOL	Masuda et al. [365]
Cattle	China	526	54 (10.3)	ST4, ST5, ST10, ST14	MOL	Zhu et al. [366]
Cattle	China	147	14 (9.5)	ST3, ST10, ST14	MOL	Wang et al. [367]
Cattle	China	57	15 (26.3)	ST10, ST14	MOL	Zhang et al. [14]
Cattle	Korea	1512	101 (6.7)	ST1, ST5, ST10, ST14	MOL	Lee et al. [368]
Cattle	United Arab Emirates	22	5 (22.7)	ST10	MOL	AbuOdeh et al. [369]
Deer (Caspian red deer)	Iran	1	1 (100.0)	NA	CM	Mirzapour et al. [370]
Deer (Javan rusa)	Malaysia	50	14 (28.0)	ST10	MOL	Mohammad et al. [371]
Deer (Mousedeer)	Malaysia	4	1 (25.0)	Unknown (Clade IV)	IVC, MOL	Mohd Zain et al. [372]
Deer (Sambar deer)	Malaysia	14	4 (28.6)	NA	IVC	Hemalatha et al. [356]
Deer (Sika deer)	Malaysia	50	16 (32.0)	ST10	MOL	Mohammad et al. [371]
Deer (Red deer)	China	5	2 (40.0)	ST10	MOL	Li et al. [353]
Deer (Red deer/Wapiti)	China	3	1 (33.3)	ST10	MOL	Zhao et al. [352]
Deer (Reindeer)	China	104	7 (6.7)	ST10, ST13	MOL	Wang et al. [373]
Deer (Fallow deer)	China	2	1 (50.0)	ST10	MOL	Zhao et al. [352]
Deer (White-lipped deer)	China	1	1 (100.0)	ST10	MOL	Zhao et al. [352]
Deer (Sika deer)	China	8	3 (37.5)	ST10	MOL	Zhao et al. [352]
Deer (Sika deer)	China	82	12 (14.6)	ST10, ST14	MOL	Wang et al. [373]
Deer (Sika deer)	China	11	1 (9.1)	ST1	MOL	Deng et al. [3]
Deer (Sika deer)	China	760	6 (0.8)	ST10, ST14	MOL	Ni et al. [374]
Deer (Spotted deer)	Bangladesh	30	1 (3.3)	ST14	MOL	Li et al. [375]
Deer (Water deer)	Korea	125	51 (40.8)	ST4, ST14	MOL	Kim et al. [376]
Eland	China	9	6 (66.7)	ST10, ST14	MOL	Zhao et al. [352]
Gayal	Bangladesh	4	1 (25.0)	ST14	MOL	Li et al. [375]
Giraffe	China	10	2 (20.0)	ST12	MOL	Zhao et al. [352]
Goat	China	789	458 (58.0)	ST1, ST3, ST4, ST5, ST10, ST14	MOL	Song et al. [377]
Goat	China	781	2 (0.3)	ST1	MOL	Li et al. [378]
Goat	China	59	28 (47.5)	ST10, ST14	MOL	Zhang et al. [14]
Goat	Nepal	400	3 (0.8)	NA	CM	Ghimire and Bhattarai [379]
Goat	Malaysia	236	73 (30.9)	ST1, ST3, ST6, ST7	MOL	Tan et al. [380]
Goat	Malaysia	31	8 (25.8)	ST4, ST8, ST10	MOL	Noradilah et al. [15]
Goat	Malaysia	65	14 (21.5)	NA	IVC	Abd Razak et al. [357]
Goat	Malaysia	20	13 (65.0)	NA	IVC	Hemalatha et al. [356]
Goat	Nepal	29	1 (3.4)	ST4	IVC, MOL	Lee et al. [18]
Goat	Philippines	6	1 (16.7)	ST14	IVC, MOL	Adao et al. [381]
Goat	Thailand	38	36 (94.7)	ST10, ST12, ST14	MOL	Udonsom et al. [215]
Goral (Himalayan)	Nepal	19	1 (5.3)	NA	CM	Adhikari et al. [382]
Guanaco	China	20	14 (70.0)	ST10, ST22	MOL	Zhao et al. [352]
Guar	Malaysia	10	3 (30.0)	NA	IVC	Hemalatha et al. [356]
Oryx	China	2	1 (50.0)	ST10	MOL	Zhao et al. [352]
Oryx	China	11	1 (9.1)	ST5	MOL	Li et al. [353]
Pig	Cambodia	73	33 (45.2)	ST5	MOL	Wang et al. [242]
Pig	China	560	419 (74.8)	ST1, ST3, ST5, ST10	MOL	Song et al. [383]
Pig	China	68	6 (8.8)	ST5	MOL	Wang et al. [367]
Pig	China	801	174 (21.7)	ST1, ST3, ST5	MOL	Wang et al. [384]
Pig	China	866	433 (50.0)	ST1, ST3, ST5	MOL	Han et al. [385]
Pig	China	396	170 (42.9)	ST1, ST5	MOL	Zou et al. [386]
Pig	India	1	1 (100.0)	NA	CM	Sreekumar et al. [355]
Pig	India	90	85 (94.4)	NA	CM	Arpitha et al. [387]
Pig	Indonesia	93	81 (87.1)	ST1, ST2, ST5, ST7	IVC, MOL	Yoshikawa et al. [41]
Pig	Indonesia	100	63 (63.0)	NA	CM	Mahendra et.al. [388]
Pig	Indonesia	100	69 (69.0)	NA	CM	Widisuputri et al. [389]
Pig	Korea	646	390 (60.4)	ST1, ST2, ST3, ST5	MOL	Paik et al. [390]
Pig	Nepal	11	4 (36.4)	ST4	IVC, MOL	Lee et al. [18]
Pig	Philippines	49	36 (73.5)	ST1, ST2, ST3, ST5	MOL	Adao et al. [391]
Pig	Philippines	99	20 (20.2)	ST1, ST5, ST7	IVC, MOL	Adao et al. [381]
Pig	Philippines	122	47 (38.5)	NA	CM, IVC	De La Cruz et al. [392]
Pig	Philippines	100	14 (14.0)	ST1, ST5	IVC, MOL	Evidor and Rivera [393]
Pig	Philippines	101	2 (2.0)	NA	CM	Murao et al. [394]
Pig	Thailand	102	32 (31.4)	ST1, ST3, ST12, ST14	MOL	Sanyanusin et al. [395]
Pig	Thailand	90	32 (35.6)	ST1, ST3, ST5	MOL	Pintong et al. [214]
Pig	Thailand	87	40 (46.0)	ST1, ST5	MOL	Udonsom et al. [215]
Pig	Malaysia	10	10 (100.0)	NA	IVC	Hemalatha et al. [356]
Pig	Vietnam	12	12 (100.0)	ST5	MOL	Alfellani et al. [396]
Sheep	Iran	150	29 (19.3)	ST7, ST10	CM, MOL	Rostami et al. [364]
Sheep	China	832	50 (6.0)	ST5, ST10, ST14	MOL	Li et al. [378]
Sheep	China	109	6 (5.5)	ST1, ST5, ST10, ST14	MOL	Wang et al. [367]
Sheep	China	38	16 (42.1)	ST2, ST10, ST14	MOL	Zhang et al. [14]
Sheep	China	78	42 (53.8)	ST2, ST10, ST14	MOL	Zhang et al. [14]
Sheep	United Arab Emirates	11	7 (63.6)	ST10, ST14	MOL	AbuOdeh et al. [369]
Sheep	Malaysia	38	22 (57.9)	NA	IVC	Hemalatha et al. [356]
Sheep	Malaysia	20	2 (10.0)	NA	IVC	Abd Razak et al. [357]
Small ruminants	India	107	15 (14.0)	NA	CM	Arpitha et al. [387]
Takin	China	49	28 (57.1)	ST10, ST12, ST14	MOL	Zhao et al. [352]
Waterbuck	China	3	3 (100.0)	ST12, ST14	MOL	Zhao et al. [352]
Waterbuck	China	2	1 (50.0)	ST21	MOL	Zhao et al. [352]
Waterbuck	Bangladesh	7	1 (14.3)	ST10	MOL	Li et al. [375]
Wild boar	South Korea	433	45 (10.4)	ST5	MOL	Lee et al. [397]
Wild Boar	Iran	25	11 (44.0)	NA	CM	Yaghoobi et al. [398]
Wild Boar	Iran	1	1 (100.0)	NA	CM	Mirzapour et al. [370]
Yak	China	1027	278 (27.1)	ST10, ST12, ST14	MOL	Ren et al. [399]
Yak	China	102	39 (38.2)	ST2, ST10, ST14	MOL	Zhang et al. [14]
Yak	China	6	3 (50.0)	ST10, ST14	MOL	Zhao et al. [352]
**Perissodactyla**						
Horse	China	32	9 (28.1)	ST2, ST10	MOL	Zhang et al. [14]
Horse	China	4	1 (25.0)	ST10	MOL	Zhao et al. [352]
Wild Ass	China	5	2 (40.0)	ST10, ST12	MOL	Zhao et al. [352]
Pony	China	6	1 (16.7)	ST10	MOL	Zhao et al. [352]
Zebra	China	7	1 (14.3)	ST10	MOL	Li et al. [353]
**Proboscidea**						
Elephant	Bangladesh	3	1 (33.3)	ST11	MOL	Li et al. [375]

CM—Conventional microscopy, IVC—In vitro cultivation, MOL—Molecular technique, NA—Not applicable.

**Table 10 biology-10-00990-t010:** Prevalence and subtype distribution of *Blastocystis* spp. in carnivorous animals in Asia (2010–2021).

Host	Country	No. of Samples Examined	Number of Positive Samples (%)	Subtypes (STs) Identified	Method(s)	References
Artic fox	China	213	4 (1.9)	ST1, ST4, ST7	MOL	Wang et al. [373]
Bear	China	12	3 (25.0)	ST17	MOL	Deng et al. [3]
Bear	China	312	45 (14.4)	ST1	MOL	Ni et al. [374]
Cat	China	346	2 (0.6)	ST1	MOL	Li et al. [400]
Cat	Indonesia	90	48 (53.3)	NA	MOL	Patagi et al. [401]
Cat	Iran	140	20 (14.3)	NA	CM	Khademvatan et al. [402]
Cat	Iran	119	21 (17.7)	ST1, ST3, ST4, ST10, ST14	MOL	Mohammadpour et al. [403]
Cat	South Korea	158	1 (0.6)	ST4	MOL	Kwak and Seo [404]
Cat	Malaysia	60	12 (20.0)	ST1	MOL	Farah Haziqah et al. [405]
Cat	Turkey	3	3 (100.0)	ST3	MOL	Eroglu and Koltas [19]
Common raccoon	Iran	30	5 (6.7)	ST1, ST2, ST3	MOL	Mohammad Rahimi et al. [406]
Dog	China	136	4 (2.9)	ST1, ST4	MOL	Wang et al. [373]
Dog	China	651	35 (5.4)	ST1, ST3, ST10	MOL	Liao et al. [407]
Dog	India	80	19 (24.0)	ST1, ST4, ST5, ST6	MOL	Wang et al. [408]
Dog	Iran	301	59 (19.6)	NA	CM	Mohaghegh et al. [409]
Dog	Iran	552	29 (5.2)	NA	CM	Mirbadie et al. [410]
Dog	Iran	154	29 (18.8)	ST2, ST3, ST4, ST7, ST8, ST10	MOL	Mohammadpour et al. [403]
Dog	Turkey	4	4 (100.0)	ST1, ST2	MOL	Eroglu and Koltas [19]
Dog	Philippines	145	21 (14.5)	ST1, ST2, ST3, ST4, ST5	IVC, MOL	Belleza et al. [297]
Dog	Malaysia	84	40 (47.6)	ST1, ST3, ST4, ST8, ST10	MOL	Noradilah et al. [15]
Dog	Thailand	13	1 (7.7)	ST3	MOL	Udonsom et al. [215]
Dog	Cambodia	80	1 (1.3)	ST2	MOL	Wang et al. [408]
Dog	China	7	1 (14.3)	ST10	MOL	Li et al. [353]
Leopard	China	3	2 (66.7)	ST1, ST5	MOL	Deng et al. [3]
Meerkat	Iran	1	1 (100.0)	NA	CM	Mirzapour et al. [370]
Meerkat	China	2	1 (50.0)	ST5	MOL	Li et al. [353]
Panda (Giant panda)	China	81	10 (12.3)	ST1	MOL	Deng et al. [411]
Panda (Red panda)	China	23	2 (8.7)	ST1	MOL	Deng et al. [411]
Raccoon dog	China	40	3 (7.5)	ST3	MOL	Wang et al. [373]
Tiger (Siberian tiger)	China	13	1 (7.7)	ST10	MOL	Li et al. [353]
Tiger (White tiger)	China	9	1 (11.1)	ST10	MOL	Li et al. [353]

CM—Conventional microscopy, IVC—In vitro cultivation, MOL—Molecular technique, NA—Not applicable.

**Table 11 biology-10-00990-t011:** Prevalence and subtype distribution of *Blastocystis* spp. in non-human primates in Asia (2010–2021).

Host	Country	No. of Samples Examined	Number of Positive Samples (%)	Subtypes (STs) Identified	Method(s)	References
**Primates**						
Langur	Bangladesh	5	3 (60.0)	ST1, ST13	MOL	Li et al. [375]
Grey langur	Bangladesh	2	1 (50.0)	ST1	MOL	Li et al. [375]
White-cheeked gibbon	China	4	1 (25.0)	ST1	MOL	Ma et al. [250]
White-cheeked gibbon	China	4	4 (100.0)	ST2, ST3	MOL	Deng et al. [3]
Ring-tailed lemur	China	6	2 (33.3)	ST2, ST4	MOL	Li et al. [353]
Ring-tailed lemur	China	16	7 (43.8)	ST3, ST5, ST9	MOL	Ma et al. [250]
Ring-tailed lemur	China	13	6 (46.2)	ST1, ST2	MOL	Deng et al. [3]
Macaque	China	97	85 (87.6)	ST1, ST2, ST3, ST5, ST7	MOL	Zanzani et al. [412]
Macaque	China	185	12 (7.0)	ST1, ST2, ST3	MOL	Zhu et al. [173]
Macaque (experimental)	China	505	235 (46.5)	ST1, ST2, ST3	MOL	Li et al. [413]
Rhesus macaque	Bangladesh	62	20 (32.3)	ST1, ST2, ST3	MOL	Li et al. [375]
Rhesus macaque	China	29	28 (96.6)	ST1, ST2, ST3, ST19	MOL	Zhao et al. [352]
Rhesus macaque	China	17	10 (58.8)	ST1	MOL	Deng et al. [3]
Rhesus macaque	China	18	6 (33.3)	ST2, ST3	MOL	Ma et al. [250]
Japanese macaque	China	33	6 (18.2)	ST2, ST3	MOL	Ma et al. [250]
Macaque	Philippines	50	5 (10.0)	NA	CM	Casim et al. [414]
Long-tailed macaque	Thailand	628	263 (41.9)	ST1, ST2, ST3	IVC, MOL	Vaisusuk et al. [415]
Crab-eating macaque	China	13	3 (23.1)	ST2, ST3	MOL	Ma et al. [250]
Orangutan	Indonesia	262	36 (13.7)	NA	CM	Labes et al. [416]
Orangutan	Malaysia	10	5 (50.0)	NA	IVC	Hemalatha et al. [356]
Vervet monkey	Iran	40	3 (7.5)	NA	CM	Dalimi et al. [417]
Vervet monkey	Bangladesh	7	3 (42.9)	ST2, ST3, ST13	MOL	Li et al. [375]
Hamadryas baboon	Saudi Arabia	823	349 (42.4)	NA	CM	Alqumber [303]
Hamadryas baboon	China	23	13 (56.5)	ST1, ST3	MOL	Zhao et al. [352]
Chimpanzee	China	10	8 (80.0)	ST2	MOL	Zhao et al. [352]
Chimpanzee	China	15	3 (13.3)	ST1, ST5	MOL	Ma et al. [250]
Francois’ leaf monkey	China	1	1 (100.0)	ST2	MOL	Zhao et al. [352]
Francois’ leaf monkey	China	3	2 (66.7)	ST1	MOL	Ma et al. [250]
Mandrill	China	4	1 (25.0)	ST3	MOL	Zhao et al. [352]
Mandrill	China	15	9 (60.0)	ST1, ST4	MOL	Ma et al. [250]
De Brazza’s monkey	China	5	4 (80.0)	ST1, ST10	MOL	Zhao et al. [352]
De Brazza’s monkey	China	5	5 (100.0)	ST1, ST2	MOL	Ma et al. [250]
Golden snub-nosed monkey	China	46	41 (89.1)	ST1, ST13	MOL	Zhao et al. [352]
Snub-nosed monkey	China	22	9 (40.9)	ST1, ST2	MOL	Ma et al. [250]
Golden monkey	China	37	18 (48.6)	ST1, ST2, ST3	MOL	Ma et al. [418]
Squirrel monkey	China	93	19 (20.4)	ST17	MOL	Deng et al. [3]
Common squirrel monkey	China	30	9 (30.0)	ST1, ST5	MOL	Ma et al. [250]
Red-faced spider monkey	China	4	2 (50.0)	ST2, ST3	MOL	Ma et al. [250]
Monkey	Philippines	4	4 (100.0)	ST1, ST2, ST3	MOL	Rivera [21]
Non-human primates	Malaysia	308	5 (1.6)	NA	CM	Adrus et al. [419]

CM—Conventional microscopy, IVC—In vitro cultivation, MOL—Molecular technique, NA—Not applicable.

**Table 12 biology-10-00990-t012:** Prevalence and subtype distribution of *Blastocystis* spp. in birds in Asia (2010–2021).

Host	Country	No. of Samples Examined	Number of Positive Samples (%)	Subtypes (STs) Identified	Method(s)	References
Duck	Philippines	31	3 (9.6)	ST7, *B. pythoni*	IVC, MOL	Adao et al. [381]
Birds	Turkey	5	5 (100.0)	ST1, ST2	MOL	Eroglu and Koltas [19]
Chicken	China	46	6 (13.0)	ST6, ST7	MOL	Wang et al. [373]
Chicken	Philippines	34	5 (14.7)	ST7, Mixed	IVC, MOL	Adao et al. [381]
Chicken	India	24	20 (83.3)	NA	CM	Sreekumar et al. [355]
Chicken	India	170	50 (29.4)	NA	CM	Arpitha et al. [387]
Chicken	Indonesia	38	13 (34.2)	ST7	IVC, MOL	Yoshikawa et al. [41]
Chicken	Lebanon	223	71 (31.8)	ST6, ST7	MOL	Greige et al. [199]
Chicken	Malaysia	104	27 (26.0)	ST1, ST3, ST6, ST7, ST9	MOL	Noradilah et al. [15]
Chicken	Malaysia	15	1 (6.7)	ST6	MOL	Mohammad et al. [288]
Chicken	Malaysia	107	27 (25.2)	NA	IVC	Farah Haziqah et al. [420]
Chicken	Malaysia	179	47 (26.3)	ST1, ST6, ST7, ST8	IVC, MOL	Farah Haziqah et al. [421]
Crested ibis	China	63	6 (9.5)	NA	CM	Zhang et al. [422]
Crow (Hooded)	Iran	144	64 (44.4)	ST13, ST14	IVC, MOL	Asghari et al. [423]
Duck	Malaysia	20	8 (40.0)	ST1, ST2, ST3, ST7	MOL	Noradilah et al. [15]
Green-naped lorikeet	China	2	1 (50.0)	ST10	MOL	Li et al. [353]
Ostrich	China	9	3 (33.3)	ST5, ST10, ST20	MOL	Zhao et al. [352]
Ostrich	China	19	6 (31.6)	ST5	MOL	Deng et al. [3]
Ostrich	Malaysia	37	37 (100.0)	ST6	IVC, MOL	Chandrasekaran et al. [424]
Ostrich	Malaysia	37	37 (100.0)	NA	IVC	Hemalatha et al. [424]
Ostrich	China	3	2 (66.7)	ST5	MOL	Li et al. [353]
Green peafowl	China	12	1 (8.3)	ST3	MOL	Deng et al. [3]
Green peafowl	China	15	1 (6.7)	ST8	MOL	Deng et al. [411]
Indian peafowl	China	20	3 (15.0)	ST7, ST8	MOL	Li et al. [353]
Pigeon	China	34	4 (11.8)	ST8	MOL	Deng et al. [3]
Pigeon	China	47	1 (2.1)	ST6	MOL	Wang et al. [373]
Pigeon	Iran	156	67 (42.9)	ST13	IVC, MOL	Asghari et al. [423]
Poultry	Iran	132	21 (15.9)	ST7, ST10, ST14	CM, MOL	Rostami et al. [364]
Red crowned crane	China	43	6 (14.0)	ST6, ST7	MOL	Wang et al. [373]
Red-crowned crane	China	2	1 (50.0)	ST14	MOL	Li et al. [353]
Ruddy shelduck	China	11	2 (18.2)	ST8	MOL	Deng et al. [411]
Swan	Malaysia	20	7 (35.0)	ST1, ST3	MOL	Noradilah et al. [15]
Black swan	China	38	4 (10.5)	ST8	MOL	Deng et al. [411]
Turkey	India	4	3 (75.0)	NA	CM	Sreekumar et al. [355]

CM—Conventional microscopy, IVC—In vitro cultivation, MOL—Molecular technique, NA—Not applicable.

**Table 13 biology-10-00990-t013:** Prevalence and subtype distribution of *Blastocystis* spp. in rodents in Asia (2010–2021).

Host	Country	No. of Samples Examined	Number of Positive Samples (%)	Subtypes (STs) Identified	Method(s)	References
Flying squirrel	China	207	63 (30.4)	ST1, ST3, ST13	MOL	Xiao et al. [425]
Eastern chipmunk	China	171	8 (4.7)	ST4	MOL	Chai et al. [426]
Eurasian red squirrel	China	72	7 (9.7)	ST4	MOL	Chai et al. [426]
Black great squirrel	China	1	1 (100.0)	ST4	MOL	Deng et al. [3]
Red giant flying squirrel	China	1	1 (100.0)	ST4	MOL	Deng et al. [3]
Indian palm squirrel	United Arab Emirates	4	2 (50.0)	ST4	MOL	AbuOdeh et al. [369]
Shrew-faced squirrel	United Arab Emirates	1	1 (100.0)	ST17	MOL	AbuOdeh et al. [369]
Chinese striped hamster	China	98	12 (12.2)	ST4	MOL	Chai et al. [426]
Chinchilla	China	72	3 (4.2)	ST4, ST17	MOL	Chai et al. [426]
Chinchilla	China	6	4 (66.7)	ST17	MOL	Deng et al. [3]
Guinea pig	China	90	12 (13.3)	ST4	MOL	Chai et al. [426]
Patagonian mara	China	15	3 (20.0)	ST4	MOL	Li et al. [353]
Rat *(Mus musculus)*	China	108	4 (3.7)	ST4	MOL	Wang et al. [373]
Laboratory rats	China	355	29 (8.2)	ST4, ST7	MOL	Li et al. [427]
Rat *(Rattus exulans)*	Indonesia	77	10 (13.0)	ST4	IVC, MOL	Yoshikawa et al. [41]
Rat	Indonesia	98	6 (6.0)	NA	CM	Prasetyo [428]
Rat *(Rattus exulans*)	Indonesia	67	11 (16.4)	ST4	MOL	Katsumata et al. [429]
Rodents	Iran	52	3 (5.8)	NA	CM	Seifollahi et al. [430]
Rat *(Rattus norvegicus)*	Iran	127	20 (15.8)	ST1, ST3, ST4	MOL	Mohammadpour et al. [403]
Rat *(Rattus norvegicus)*	Malaysia	95	48 (51.0)	NA	CM	Premaalatha et al. [431]
Rat *(Rattus norvegicus)*	Malaysia	290	133 (45.9)	ST1, ST4, ST5, ST7	IVC, MOL	Farah Haziqah et al. [432]
Wild rats (*Rattus novercious*)	Japan	48	12 (25.0)	ST4	MOL	Katsumata et al. [429]
Swiss-Webster mice	Iran	50	1 (2.0)	NA	CM	Kalani et al. [433]

CM—Conventional microscopy, IVC—In vitro cultivation, MOL—Molecular technique, NA—Not applicable.

**Table 14 biology-10-00990-t014:** Prevalence and subtype distribution of *Blastocystis* spp. in reptiles in Asia (2010–2021).

Host	Country	No. of Samples Examined	Number of Positive Samples (%)	Subtypes (STs) Identified	Method(s)	References
**Squamata**						
Cobra snake	Iran	1	1 (100.0)	NA	CM	Mirzapour et al. [370]
Albino python	Iran	1	1 (100.0)	NA	CM	Mirzapour et al. [370]
Water monitor lizard	Malaysia	6	1 (1.6)	Unknown (Clade VIII)	IVC, MOL	Mohd Zain et al. [372]
**Testudines**						
African spurred tortoise	United Arab Emirates	19	5 (26.3)	Unknown	MOL	AbuOdeh et al. [369]
Greek tortoise	United Arab Emirates	2	1 (50.0)	Unknown	MOL	AbuOdeh et al. [369]
Iguana	United Arab Emirates	1	1 (100.0)	Unknown	MOL	AbuOdeh et al. [369]

CM—Conventional microscopy, IVC—In vitro cultivation, MOL—Molecular technique, NA—Not applicable.

**Table 15 biology-10-00990-t015:** Prevalence and subtype distribution of *Blastocystis* spp. in insects and other animal groups in Asia (2010–2021).

Host	Country	No. of Samples Examined	Number of Positive Samples (%)	Subtypes (STs) Identified	Method(s)	References
**Blattodea**						
Cockroach	China	116	96 (82.8)	ST2	MOL	Ma et al. [418]
Cockroach	Thailand	920	9 (1.0)	NA	CM	Chamavit et al. [434]
Cockroach	Thailand	450	18 (4.0)	NA	CM	Dokmaikaw and Suntaravitun [435]
Cockroach *(Blatella germanica)*	Turkey	138	57 (41.0)	NA	CM	Oguz et al. [436]
Cockroach *(Blatella germanica)*	Iran	496	5 (1.0)	NA	CM	Motevalli-Haghi et al. [437]
Cockroach (*Periplaneta americana)*	Malaysia	151	61 (40.4)	ST3	IVC, MOL	Farah Haziqah et al. [438]
**Diprotodontia**						
Gray kangaroo	China	11	8 (72.7)	ST10	MOL	Zhao et al. [352]
Red-necked wallaby	China	15	2 (13.3)	ST11	MOL	Li et al. [353]
Sugar glider	Indonesia	100	100 (100.0)	NA	CM, IVC	Natalia et al. [439]
**Lagomorpha**						
New Zealand white rabbit	China	215	7 (3.3)	ST4	MOL	Wang et al. [373]
Rabbit	China	616	6 (1.0)	NA	MOL	Li et al. [440]
Rabbit	United Arab Emirates	3	1 (33.3)	ST14	MOL	AbuOdeh et al. [369]
**Eulipotyphla**						
Hedgehog	Iran	1	1 (100.0)	NA	CM	Mirzapour et al. [370]

CM—Conventional microscopy, IVC—In vitro cultivation, MOL—Molecular technique, NA—Not applicable.

**Table 16 biology-10-00990-t016:** Prevalence and subtype distribution of *Blastocystis* spp. in food and environmental sources in Asia (2010–2021).

Country	Food/Environmental Source	No. of Samples Examined	No. of Positive Samples (%)	Subtypes (STs) Identified	Method(s)	References
Iran	Treated wastewater	12	5 (41.7)	ST2, ST6, ST8	F, MOL	Javanmard et al. [441]
Malaysia	River water	480	133 (27.7)	NA	MB, IVC	Ithoi et al. [442]
Malaysia	Drinking water treatment plants	85	22 (25.9)	NA	IMS, CM	Richard et al. [443]
Malaysia	River water	14	14 (100.0)	ST1, ST2, ST3	MF, MOL	Noradilah et al. [444]
	Various water sources			ST1, ST2, ST3, ST4, ST8, ST10		
Malaysia	River water	7	3 (42.9)	NA	MF, IVC	Noradilah et al. [23]
	Village water sources	16	1 (6.3)			
Nepal	River water	4	4 (100.0)	ST1, ST4	C, MOL	Lee et al. [18]
Philippines	Wastewater (influent)	31	7 (23.0)	ST1, ST2	C, IVC, MOL	Banaticla and Rivera, [445]
	Wastewater (effluent)	31	2 (7.0)	ST1, ST2		
Turkey	Tap water	25	3 (12.0)	ST1	MOL	Eroglu and Koltas, [19]
Turkey	Streams and drinking water	228	47 (20.6)	NA	CM	Karaman et al. [446]
Turkey	River water	195	9 (4.6)	ST1, ST3	C, MOL	Koloren et al. [447]
	Sea water	48	1 (2.1)	ST1		
Turkey	Surface water	75	4 (5.3)	ST1, ST3	C, MOL	Kolören and Karaman [448]
Saudi Arabia	Leafy vegetables	470	13 (2.8)	NA	S, CM	Al-Megrin [27]
Iran	Fresh vegetables	240	10 (4.2)	NA	S, CM	Isazadeh et al. [449]
Syria	Fresh vegetables	128	13 (10.2)	NA	MOL	Al Nahhas and Aboualchamat [450]
Korea	Ambient air	71	1 (1.4)	NA	MOL	Han et al. [451]

C—Centrifugation, CM—Conventional microscopy, F—Filtration, IMS—Immunomagnetic separation technique, IVC—In vitro cultivation, MB—Membrane filtration, MOL—Molecular technique, S—Sedimentation, NA—Not applicable.

## Data Availability

All data generated or analyzed during this study are included in this published article.

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
