# Peer review of "The Coexistence of Blastocystis spp. in Humans, Animals and Environmental Sources from 2010–2021 in Asia"

_biology, 2021, doi:10.3390/biology10100990_

Round 1

Reviewer 1 Report

The research: The Coexistence of Blastocystis spp. in Humans, Animals and Environmental Sources from 2010 – 2021 in Asia by Adedolapo Aminat Rauff-Adedotun, Farah Haziqah Meor Termizi, Nurshafarina Shaari and II LI Lee is very interesting and gives new information.

Comments on the paper:

Title reflects the paper’s content.

Abstract is appropriate.

Introduction

The introduction makes a proper introduction to the subject matter of the paper.

Materials and Methods, Results and Discussion

Well written.

Editoral comments:

Line 51 should be [1-3]

Line 62 should be sall-subunit ribosomal RNA (SSU) gene of…

Line 63 should be [10-11]

Line 66 should be the SSU gene

Line 71 should be [13-14]

Line 76 should be [9,14,16]

Line 77 should be [17-27]

Line 88 should be [29-30]

Reviewer 2 Report

This is an extensive review on Blastocystis spp. in Asia, covering not only humans and animals but environment as well. Because the article is well organized and heavily referenced, I believe this deserves publication in Biology.

Reviewer 3 Report

The review paper here presented is interesting and offers a significant recollection of information regarding the spread of Blastocystis spp. in Asia. Proper literature was reported. However, prior to considering this manuscript for publication in Biology, I would suggest the authors to introduce the following minor changes:

Table 1: There is an error in the first row (“Bangladesh”), where the number of positive samples is higher than the total samples considered. Also in the second column values are not reported correctly.

Table 1: Which were the conditions of the children involved in these studies? Since the variation of the % of Blastocystis within a country appears wide from the different studies (e.g., for Indonesia % ranges from 6.8 to almost 70), it would be more informative to describe in the text the clinical conditions of the subjects involved, if the information is available.

Table 2: Would be even more informative to divide the “cancer” data in different sub-categories depending on cancer type (if the information is available). Novel information should be discussed in the manuscript.

Table 2: it is not clear if the subjects considered in the studies here reported were under pharmacological treatment or not. The information should be discussed in the manuscript.

Table 5: Since pathological conditions of patients could not be reported, the data of this table are quite generic

Tables 9-15: I think that references in which only 1 or 2 animals were considered should not be included, since the statistical power here is null.
